# Cold storage-mediated rearing of *Trichogramma evanescens* Westwood on eggs of *Plodia interpunctella* (Hübner) and *Galleria mellonella* L.

**Aslam Haque[1], Saiful Islam[1], Abdul Bari[2], Akhtar Hossain[3], Christos G. Athanassiou[4], Mahbub Hasan**[ORCID][3]*

**1** Department of Crop Science and Technology, Rajshahi University, Rajshahi, Bangladesh, **2** Insect Biotechnology Division, Institute of Food and Radiation Biology, Atomic Energy Research Establishment, Savar, Dhaka, Bangladesh, **3** Department of Zoology, Rajshahi University, Rajshahi, Bangladesh, **4** Laboratory of Entomology and Agricultural Zoology, Department of Agriculture, Crop Production and Rural Environment, University of Thessaly, N. Ionia Magnesia, Greece

* mmhbgd@yahoo.com

**Data Availability Statement:** All relevant data are within the paper.

## Abstract

The egg parasitoid *Trichogramma evanescens* Westwood is considered as an efficient biological control agent for managing several lepidopteran pests and it is widely distributed throughout the world. Mass rearing protocols of parasitoids that are currently in use in biocontrol programs require a meticulous quality control plan, in order to optimize their efficacy, but also their progeny production capacity. In this paper, the effect of different factors on the quality control in mass rearing of *T. evenescens*, using *Plodia interpunctella* (Hübner) and *Galleria mellonella* L. as host species, were investigated. The impact of egg agewas significant in the rates of parasitism, for both host species tested. Significantly highest percent of parasitoid emergence was noticed in two day-old eggs for both host species, while one day-old eggs day exhibited the maximum emergence when both species were used togetherin the same trials. Age-dependent storage egg preservation at either 4 or 9°C significantly influenced the parasitism percentages on both species. The highest parasitism percentage was recorded in two day-old *G. mellonella* eggs that are kept for 15 days at 9°C while the lower in one day-old *P. interpunctella* eggs for 60 d storage. Moreover, the highest parasitoid mortality was recorded in *T. evanescens* reared either on *P. interpunctella* or *G. mellonella* at 20°C. Rearing of the parasitoid on a mixture of eggs of both host species resulted in higher parasitism, but not always in higher rates of parasitoid emergence. The results of the present work provide useful information that can be further utilized in rearing protocols of *T. evanescens*.

## Introduction

The egg parasitoids of the genus *Trichogramma* (Hymenoptera: Trichogrammatidae) have been extensively used as efficient biocontrol agents in integrated pest management programs

**Funding:** The author(s) received no specific funding for this work.

**Competing interests:** The authors have declared that no competing interests exist.

throughout the world [1–7]. Among the factors that determine the efficacy of *Trichogramma* spp. is host egg age [8,9]. The parasitoids usually exhibit a significant preference for young host eggs, while in some cases females do not parasitize old eggs at all [10,11]. For instance, Abd El-Hafez [12] reported that *Trichogramma evanescens* Westwood and *T. bactrae* Nagaraja females preferred to oviposit into young eggs of the pink bollworm, *Pectinophora gossypiella* (Saunders) (Lepidoptera: Gelechiidae) and the spiny bollworm, *Earias biplaga* Walger (Lepidoptera: Nolidae), as compared to old eggs of these two species. The age of host eggs is mainly utilized through two indicators in the production of *Trichogramma* spp. progeny: firstly, the oviposition preference of the parasitoid females and secondly, the resource quality available for the developing parasitoid larvae, in terms of physiological host-parasitoid interactions [13].

The continuous availability of host eggs is a key element in the mass production protocols of *Trichogramma* spp. However, in these protocols timing is essential, as the parasitized eggs may be overproduced and discarded when they are not needed to be used for field release. Therefore, the development of effective "storage" methods for *Trichogramma* spp. are of utmost importance for the successful implementation of these commercial biological control agents, as well as for the efficiency in parasitoid mass production [14,15]. In this context, cold storage techniques are being considered as a valuable tool in the rearing of parasitoids since they provide a constant supply for research and enables flexibility in mass production [15,16]. Cold storage meets with certain advantages since it facilitates the upsurge of sufficient numbers of parasitoids for future releases [16], thereby minimizing the cost of maintaining colonies when they are not required [17]. Therefore, the impact of cold storage on the performance and population growth of parasitoids has received substantial interest [18–21]. Previous studies clearly indicate that most parasitoids can be cold-stored for certain periods with minimal reduction in their fitness. Cold storage has been studied in conjunction with other parameters, such as survival, sex ratio, lifespan, reproductive potential, fitness cost and intergenerational effects [22–24].

In *Trichogramma* spp., performance is defined by several key factors, including searching behavior (habitat and host location), host preference (recognition, acceptance, suitability) and tolerance to environmental conditions [25]. In addition, the parasitoid tolerance to extreme temperatures is an important factor since it determines a species' establishment and effectiveness in augmentative biological control programs [26]. Many studies have demonstrated that temperature plays a significant role in the success of effective rearing of *Trichogramma* spp. [27,28]. Still, the interactions of temperature along with different cold storage periods, on rearing of these egg parasitoid species, in conjunction with synchronization patters with the host eggs, are poorly understood. In the current study, we used *T. evenescens* as a model species in order to demonstrate its performance on eggs of different ages belonging to the Indian meal moth, *Plodia interpunctella* (Hübner) (Lepidoptera: Pyralidae) and the wax moth, *Galleria mellonella* L. (Lepidoptera: Pyralidae), which are both common hosts of this species [29]. This was carried out in conjunction with illustrating the influence of cold storage parameters in survival and parasitism rates, towards the development of quality control attributes in parasitoid rearing.

## Material and methods

### Ethics statement

This experiment did not involve any endangered or protected species.

**Host rearing.** *Galleria mellonella* culture was originally obtained from the Post-Harvest Entomology Laboratory, Department of Zoology, Rajshahi University, and the rearing culture was developed following procedures as described by Marston et al. [30]. Similarly, *P.*

*interpunctella* individuals were also collected from the stock cultures maintained at the Post-Harvest Entomology Laboratory since 2014, and they were reared on a diet of corn meal, chick laying mash, chick starter mash, and glycerol at a volumetric ratio of 4:2:2:1, respectively [31]. Both cultures were maintained in an incubator set at 27˚C, 70% relative humidity (RH), with a photoperiod of 16:8 (L: D) h.

## Parasitoid origin and rearing

The parasitoid was originally received from the Bangladesh Agricultural Research Institute (BARI), Gazipur, Bangladesh. The species was reared using eggs(<24 h old) of *P. interpunctella*, following the method of Hegazi et al. [32]. The host eggs were stacked in a paper strip (1.5×3 cm) with gum arabic glue and placed in a transparent glass vial (3.5 cm in diameter x 12 cm in length) exposed to parasitoids [33]. The glass vials were covered by cloth-wrapped cotton. The egg strips were renewed daily to avoid super parasitism.

## Experimental procedures

**Host age-dependent parasitoid performance.** To estimate the age-dependent performance of *T. evanescens*, 1, 2 and 3d-old eggs of *P. interpunctella* and *G. mellonella* were used. A total of 25 eggs of each age and species were stacked in a paper strip with gum arabic glue and placed in a transparent glass vial, as noted above, containing one-day old male and female pair of *T. evanescens*. The host eggs of both species were exposed to the parasitoid for 24 h. The mouths of the vials were wrapped in a white filter paper with the help of rubber and then were kept separately in an incubator set at 25˚C, 16:8 (L:D) photoperiod and 70% RH. Twelve host eggs were taken for each age and individual species for carrying out the experiment of combination of host eggs. There were three replications for each age and species. The percent parasitism (%) of *T. evanescens* was recorded based on eggs that turned into blackened colour. The glass vials were checked every day until all the adult parasitoids were dead. During this period, we recorded parasitism and emergence, as well as the female: male proportion. Parasitism, emergence, adult longevity and developmental period of *T. evanescens* were recorded in the $F_1$ individuals. After death, the adult body length was also measured using ocular micrometer under the microscope.

**Cold storage impact on the parasitoids.** Two age groups, i.e. 1 and 2-old, of host eggs of *P. interpunctella* and *G. mellonella* were selected for different exposure times to low temperatures, i.e. 0 (control), 15, 30, 45 and 60 d at either 4or 9˚C [34]. Twenty-five eggs of each host species and age were stacked in a paper strip with gum arabic glue and placed in a transparent glass vial, as above. The mouths of the vials were wrapped in a white filter paper with the help of rubber. After that, they were kept separately in incubators set at either 4 or 9˚C, with the rest of the conditions were set at 16:8 (L:D) photoperiod and 70% for RH, for each of the different storage periods. The glass vials were removed after the completion of each storage period. Then a single pair of male and female *T. evanescens* was introduced separately into the vials containing the cold stored host eggs for each host age, species and storage period for parasitism and kept in an incubator set at 25˚C, 16:8 (L:D) photoperiod and 70% RH. A small cotton absorbing 10% aqueous sucrose-glucose-fructose mixture (1:1:1) was supplied in the vial for feeding the *T. evanescens* adults. After 2 days of parasitism, the host eggs were checked for the percent of parasitism. The effects of cold storage in the percentage of adult emergence and sex ratio in *T. evanescens* were recorded separately for each host species. The adult body length was also measured using an eye piece-micrometer (New York Microscope Company, Hicksville, NY, USA). There were three replicates for each cold temperature, host species, age group and storage periods.

**Temperature-dependent fitness of parasitoids.** To evaluate the temperature-dependent fitness of *T. evanescens*, 1 d-old host eggs of each species was exposed at different temperatures, i.e. 15, 20, 25 and 30˚C. Twenty-five eggs of each species and age were stacked in a paper strip with gum arabic glue and placed in a vial, as noted above. The mouths of the vials were wrapped in a white filter paper with the help of rubber. After that, a single pair of male and female *T. evanescens* was introduced separately into the vials containing host eggs for each species and kept separately in incubators set at the temperatures mentioned above, and at 16:8 (L:D) photoperiod and 70% RH. Percentage of parasitism, adult emergence, sex-ratio, developmental periods and adult body length were evaluated following the procedures described above for cold storage experiments. There were three replicates for each cold temperature, host species, age group and storage period.

**Data analysis.** The assumptions of normality and homogeneity of variance were determined using Levene's test [35]. The percentages of parasitism, adult emergence and female progeny of parasitoids for all the trials were transformed to arcsine square root for stabilizing variances before being subjected to ANOVA, but the untransformed data are presented in the results for clarity. For each of the different trials including the effects of host age, cold storage-dependent and temperature, the data were subjected to analysis by multi-factorial ANOVA with the "host egg age" and "host species" as factors using the PROC ANOVA [36]. A *k*-value was calculated for different temperatures that are primarily responsible for an increase or decrease in the number of parasitoids in a given population. Means were compared by Tukey-Kramer HSD test at the 5% level.

## Results

### Host age-dependent parasitoid performance

Parasitism rates varied significantly among the different egg age groups (F = 17.60; df = 2,24; P<0.001). The highest parasitism rate was recorded in one day eggs for both host species (Table 1). However, there were no significant variations in the species and their combination for all age groups (F = 0.16; df = 3,24; P = 0.92). Moreover, the interaction age*species was not significantly different in the percent of parasitism (F = 0.77; df = 6,24; P = 0.60). Adult emergence in *T. evanescens* varied significantly in all egg age groups (F = 5.96; df = 2,24; P = 0.007) (Table 1). The highest percentage of emergence was recorded in 2 d-old eggs for both species while 1 d-old eggs showed the maximum emergence in the combination of species (Table 1). In addition, the adult emergence did not vary significantly in the species and their combinations (F = 0.75; df = 3,24; P = 0.53), but the interaction of age*species was significant (F = 2.61; df = 6,24; P = 0.04). Percent female progeny of *T. evanescens* did not differ significantly among treatments (F = 0.68; df = 11,24; P = 0.74) and the maximum (67.41) percent of female progeny was recorded in the case of *P. interpunctella* eggs, resulting from the combination with *P. interpunctella* in the case of the 2d-old eggs (Table 1). There were no significant variations in the percent female progeny among the different egg age group (F = 0.54; df = 2,24; P = 0.59) and species including their combinations (F = 0.74; df = 3,24; P = 0.54). Adult longevity of *T. evanescens* was significantly lengthened in 1 d-old eggs, compared to 2 and 3-d old eggs (F = 22.06; df = 11,24; P<0.001) (Table 1). The longevity was found to be shorter in the case of the eggs of *P. interpunctella*, especially in 2 and 3 d-old eggs. Also, parasitoid longevity was significantly different among the egg age groups (F = 81.33; df = 2,24; P<0.001) and species combinations (F = 10.67; df = 3,24; P<0.001). The interaction of age*species also varied significantly (F = 8.00; df = 6,24; P<0.001) for adult longevity. The different egg age groups significantly influenced the developmental periods in the case of both host species (F = 262.91; df = 11,24; P<0.001) (Table 1). The highest (14 d) developmental period was recorded for the

**Table 1. Age-dependent (mean±SE) biological traits in *T. evanescens* reared on eggs of *P. interpunctella* (IMM) and *G. mellonella* (WXM), placed either alone or simultaneously.**

| Age of eggs (d) | Species | No. Eggs used | % Parasitism | % Adult emergence | Female progeny (%) | Longevity (d) | Developmental periods (d) | Body Length (mm) |
|---|---|---|---|---|---|---|---|---|
| 1 | IMM | 75 | 81.33 ± 4.81[a] | 74.67 ± 7.43[b] | 54.05 ± 4.60[a] | 4.33 ± 0.29[a] | 13.00 ± 0.00[b] | 0.24 ± 0.02[b] |
| | WXM | 75 | 84.00 ± 4.62[a] | 70.67 ± 4.81[b] | 53.77 ± 6.98[a] | 3.67 ± 0.58[a] | 13.33 ± 0.33[ab] | 0.27 ± 0.01[a] |
| | *Combined* | | | | | | | |
| | IMM + | 36 | 100.00±0.00[a] | 87.50 ± 3.41[ab] | 45.15 ± 8.67[a] | 4.00 ± 0.00[a] | 12.00 ± 0.00[c] | 0.26 ± 0.02[ab] |
| | WXM | 36 | 80.83 ± 8.86[a] | 91.67 ± 6.81[a] | 52.72 ±3.66[a] | 4.00 ± 0.00[a] | 14.00 ± 0.00[a] | 0.28 ± 0.03[a] |
| | *F(df3,8)* | | 1.09 (P = 0.41) | 5.76 (P<0.02) | 0.45 (P = 0.72) | 1.33(P = 0.33) | 25.00(P<0.001) | 4.35(P<0.009) |
| 2 | IMM | 75 | 61.33 ±17.66[a] | 96.52 ± 6.26[a] | 58.43 ± 1.76[a] | 2.67 ± 0.33[b] | 11.66 ± 0.33[b] | 0.25 ±0.02[ab] |
| | WXM | 75 | 72.00± 8.01[a] | 93.75 ± 6.26[a] | 46.05 ± 9.57[a] | 3.00 ± 0.00[b] | 13.00 ± 0.00[a] | 0.26 ± 0.01[a] |
| | *Combined* | | | | | | | |
| | IMM + | 36 | 93.33 ± 3.34[a] | 69.33 ± 13.91[a] | 67.41 ± 7.07[a] | 3.00 ± 0.00[b] | 12.00 ± 0.00[b] | 0.23 ± 0.02[b] |
| | WXM | 36 | 100.00± 0.00[a] | 54.17 ± 10.21[a] | 50.36±10.93[a] | 4.00 ± 0.00[a] | 13.00 ± 0.00[a] | 0.26 ± 0.01[ab] |
| | *F(df 3,8)* | | 0.42 (P = 0.74) | 0.71 (P = 0.57) | 1.35 (P = 0.33) | 12.00 (P<0.002) | 17.00(P<0.001) | 3.58 (P<0.02) |
| 3 | IMM | 75 | 54.67 ± 5.82[a] | 84.23 ±4.62[a] | 61.72 ± 1.06[a] | 2.00 ± 0.00[b] | 8.00 ± 0.00[b] | 0.22 ± 0.01[ab] |
| | WXM | 75 | 54.67 ± 3.82[a] | 90.79 ± 4.62[a] | 54.49 ± 8.38[a] | 3.00 ± 0.00[a] | 9.00 ± 0.00[a] | 0.23 ± 0.02[a] |
| | *Combined* | | | | | | | |
| | IMM + | 36 | 77.38 ±12.44[a] | 44.44 ± 7.36[a] | 62.78±14.81[a] | 2.00 ± 0.00[b] | 8.00 ± 0.00[b] | 0.21 ± 0.02[b] |
| | WXM | 36 | 90.48 ± 4.77[a] | 55.56 ± 2.78[a] | 50.00 ± 9.63[a] | 3.00 ± 0.00[a] | 9.00 ± 0.00[a] | 0.23 ± 0.02[a] |
| | *F(df 3,8)* | | 1.03 (P = 0.43) | 0.60 (P = 0.64) | 0.39 (0.77) | 27.34 (P<0.001) | 29.43 (0.001) | 2.78 (P<0.05) |

Within a column for each age group, means with the same letter do not differ significantly; HSD test at 0.05.

combination of G. *mellonella* at the 1 d-old eggs. The developmental periods of *T. evanescens* also showed a significant variation among the egg age groups (F = 1324.00; df = 2,24; P<0.001) and species combinations (F = 69.33; df = 3,24; P<0.001). There were also significant variations in the interaction of age*species for developmental period (F = 5.83; df = 6,24; P<0.001). The adult body length of *T. evanescens* resulting from different egg age groups varied significantly (F = 6.45; df = 11,128; P<0.001) (Table 1). The maximum body length was recorded in *T. evanescens* resulting from G. *mellonella* in all age groups. The body length also varied significantly among the age (F = 21.04; df = 2,128; P<0.001) and species combinations (F = 7.53; df = 3,128; P<0.001). The interaction age*species was not significant for adult body length (F = 1.04; df = 6,128; P = 0.40).

**Cold storage impact on the parasitoids.** Age-dependent cold storage preservation significantly influenced the parasitism in *T. evanescens* reared either on *P. interpunctella* or *G. mellonella* at 4 and 9˚C (F = 5.20; df = 1,100; P<0.025) (Figs 1 and 2). A significantly higher parasitism percentage was recorded in 2 d-old egg of *G. mellonella* (F = 35.28; df = 1,16; P<0.001) for 15 d storage period at 9˚C, and the (insignificantly) lowest in 1 d-old *P. interpunctella* eggs for 60 d storage (F = 0.10; df = 1,16; P = 0.75) (Fig 2). Moreover, temperature (F = 18.74; df = 1,100; P<0.001), age (F = 5.20; df = 1,100; P<0.001) and species (F = 26.20; df = 1,100; P<0.001) varied significantly in the percent of parasitism when the host eggs were preserved at different periods of low temperatures. Adult emergence of *T. evanescens* differed significantly among the egg groups stored at different temperature levels (F = 5.87; df = 1,100; P = 0.02) (Table 2). In contrast, species (F = 0.64; df = 1,100; P = 0.42) and age (F = 0.75; df = 1,100; P = 0.39) of host eggs did not vary significantly regarding adult emergence for all

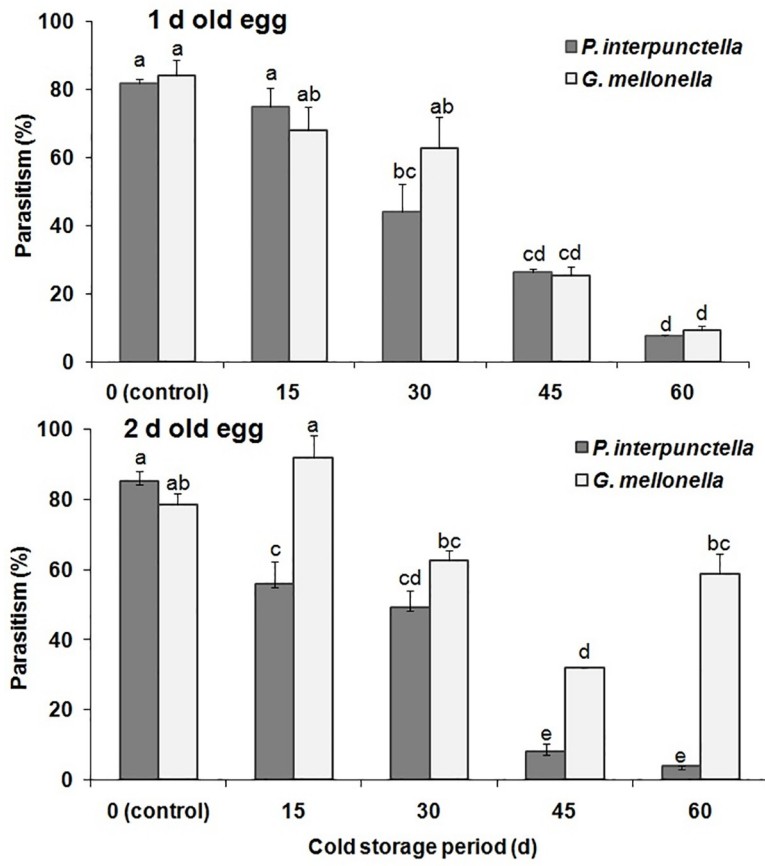

**Fig 1. Mean percentage (%±SE) of parasitism by *T. evanescens* in *P. interpunctella* and *G. mellonella* eggs of different ages on exposed to 4°C for different intervals (within each egg age and species, bars followed by the same letters are not significantly different; HSD test at 0.05).**

the storage periods, but storage period (F = 31.64; df = 4,100; P<0.001) showed a significant variation among treatments. In this context, 100% parasitoid emergence was recorded for both species, at 1 and 2 d-old eggs stored at either 4 or 9°C for 15 d (Table 2). Conversely, no adult emergence was recorded from 1d-old egg stored for 60 d at 9°C. The cold preservation of host eggs did not significantly influence the female progeny of *T. evanescens* for both species (F = 4.80; df 1,100; P = 0.03) (Table 2). The highest female progeny percent (64.21) was recorded in 2 d-old *G. mellonella* eggs stored for 30 d at 4°C. No significant effects in female progeny were recorded in the different temperatures (F = 1.96; df = 1,100; P = 0.16) and ages (F = 0.33; df = 1,100; P = 0.15), but storage periods were significant (F = 16.13; df = 4,100; P<0.001). The body length of *T. evanescens* adults resulting from the cold stored eggs varied significantly (F = 14.89; df = 4,260; P<0.001) for both temperature and species (Table 2). Increased body length was observed in the individuals coming from *G. mellonella* eggs for most storage periods. Temperature (F = 18.41; df = 1,260; P<0.001) and species (F = 13.34; df = 1,260; P = 0.001) had a significant effect in body length of the emerged adults. Nevertheless, egg age group had no significant effect in the body length of *T. evanescens* (F = 1.41; df = 1,260; P = 0.24).

**Temperature-dependent fitness of parasitoids.** Temperature insignificantly affected parasitism percentage (F = 2.04; df = 7,16; P = 0.11) with the highest percentage of parasitism

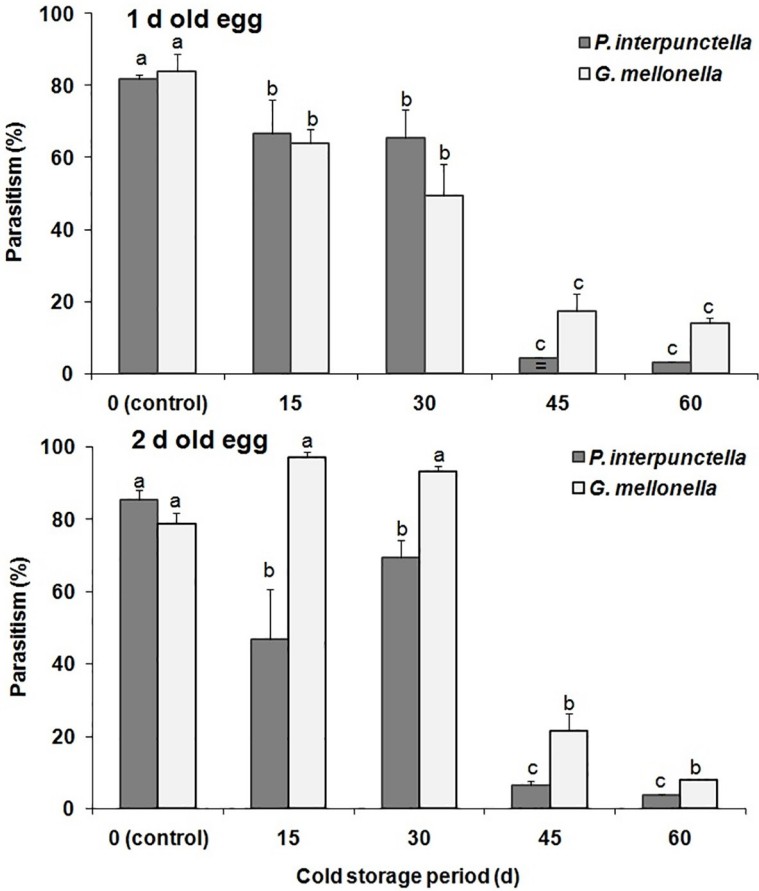

**Fig 2. Mean percentage (% ±SE) of parasitism by *T. evanescens* in *P. interpunctella* and *G. mellonella* eggs of different ages on exposed to 9˚C for different intervals (within each egg age and species, bars followed by the same letters are not significantly different; HSD test at 0.05).**

to be recorded for 25˚C in both species ([Fig 3]). For all temperatures, parasitism percentage was significantly higher in *G. mellonella* (F = 7.90; df = 3,8; P<0.008), as compared with the respective figures of *P. interpunctella* (F = 1.31; df = 3,8; P = 0.33). There were significant variations among temperatures (F = 4.52; df = 3, 16; P = 0.018), but not for species (F = 0.34; df = 1,16; P = 0.09). Moreover, the interaction between species*temperature did not also show any significant variations (F = 0.14; df = 3,16; P = 0.94). Temperature level insignificantly affected adult emergence (F = 2.67; df = 3,16; P = 0.08) ([Table 3]). The maximum levels of adult emergence were 94.9 and 100% for *P. interpunctella* and *G. mellonella*, respectively, at 25˚C. Results also showed that there were no significant variations on emergence for species (F = 1.91; df = 1,16; P = 0.19) as well as for the interaction species*temperature(F = 0.09; df = 3,16; P = 0.96). Furthermore, temperature had no significant effect in the percent female progeny of *T. evanescens* (F = 0.65; df = 3,16; P = 0.59) ([Table 3]). The highest percent of female progeny (67.14) was recorded when *T. evanescens* was reared on eggs of *P. interpunctella* at 15˚C. However, there were no significant effects on the percentage of female progeny for species (F = 1.19; df = 1, 16; P = 0.29), although the interaction species*temperature varied significantly (F = 6.94; df = 3,16; P = 0.003). The developmental periods were significantly lengthened in *T. evanescens* individuals reared either on *P. interpunctella* (F = 251.41; df = 3,8; P<0.001) or *G. mellonella* at 15˚C, with the highest period (24.67d) being recorded in *G.*

**Table 2. Mean (±SE) effect of cold storage preservation on the biological traits of *T. evanescens* reared on eggs of *P. interpunctella* (IMM) and *G. mellonella* (WXM), placed either alone or simultaneously that were maintained at either 4or 9˚C, at different intervals.**

| Temp. (˚C) | Egg age | Storage periods (d) | Species | % adult emergence | Female progeny (%) | Adult body Length(mm) |
|---|---|---|---|---|---|---|
| 4 | 1 | 0 (control) | IMM | 76.33 ± 1.45[c] | 35.75 ± 1.44[bc] | 0.27 ± 0.01[a] |
| | | | WXM | 70.67± 4.81[c] | 42.86 ±8.22[abc] | 0.28 ± 0.02[a] |
| | | 15 | IMM | 100 ± 0.00[a] | 42.74 ± 3.16[abc] | 0.21± 0.02[c] |
| | | | WXM | 100 ± 0.00[a] | 43.85 ± 8.04[abc] | 0.26 ± 0.03[a] |
| | | 30 | IMM | 97.22 ± 2.78[ab] | 48.48 ± 4.40[abc] | 0.21 ± 0.01[c] |
| | | | WXM | 98.15 ± 1.85[ab] | 56.98 + 6.20[ab] | 0.24 ± 0.02[b] |
| | | 45 | IMM | 85.00 ±2.52[abc] | 49.84 ± 3.00[abc] | 0.24 ± 0.01[b] |
| | | | WXM | 83.81 ± 1.91[bc] | 61.11 ± 5.56[a] | 0.21 ± 0.02[c] |
| | | 60 | IMM | 93.33 ± 6.67[ab] | 50.00 ± 00[abc] | 0.21± 0.01[c] |
| | | | WXM | 51.67 ± 1.67[d] | 28.16 ± 3.06[c] | 0.23±0.02[b] |
| | | *F(df = 9,20)* | | *26.50 (P<0.001)* | *3.54 (P<0.008)* | *5.63(P<0.001)* |
| | 2 | 0 (control) | IMM | 76.67 ± 2.61[bc] | 39.63 ± 4.11[ab] | 0.26 ±0.02[ab] |
| | | | WXM | 77.67 ± 1.77[ab] | 45.13 ± 4.06[ab] | 0.28 ± 0.01[a] |
| | | 15 | IMM | 93.91 ± 3.45[ab] | 43.89 ± 3.10[ab] | 0.22 ± 0.01[b] |
| | | | WXM | 95.83 ± 4.17[a] | 59.30 ± 1.74[a] | 0.25 ± 0.02[ab] |
| | | 30 | IMM | 88.21 ± 6.05b[ab] | 47.24 ± 7.87[ab] | 0.22± 0.01[b] |
| | | | WXM | 95.82 ± 2.11[a] | 64.21 ± 6.26[a] | 0.23 ± 0.03[ab] |
| | | 45 | IMM | 58.22 ± 4.82[cd] | 37.05 ± 7.59[ab] | 0.22 ±0.02[ab] |
| | | | WXM | 83.33 ± 4.17[ab] | 39.68 ± 8.85[ab] | 0.21 ± 0.02[b] |
| | | 60 | IMM | 54.67 ±0.00[d] | 26.63 ± 4.00[b] | 0.24 ±0.01[ab] |
| | | | WXM | 95.26 ± 2.48[ab] | 46.57 ± 5.20[ab] | 0.25 ± 0.01[ab] |
| | | *F (df = 9,20)* | | *16.22 (P<0.001)* | *3.55(P<0.008)* | *3.88(P<0.001)* |
| 9 | 1 | 15 | IMM | 98.41 ± 1.59[a] | 42.47 ± 9.22[ab] | 0.22 ± 0.01[b] |
| | | | WXM | 98.04 ± 1.96[a] | 62.26 ± 8.88[a] | 0.25 ± 0.02[a] |
| | | 30 | IMM | 86.54 ± 7.29[ab] | 61.24 ± 3.56[a] | 0.22 ± 0.03[b] |
| | | | WXM | 97.78 ± 2.22[a] | 43.83 ± 2.06[ab] | 0.23 ± 0.01[b] |
| | | 45 | IMM | 79.00± 2.08[b] | 40.94 ± 1.88[ab] | 0.22 ± 0.03[b] |
| | | | WXM | 98.33 ± 1.67[a] | 62.04 ± 3.21[a] | 0.21 ± 0.02[b] |
| | | 60 | IMM | - | - | - |
| | | | WXM | - | - | - |
| | | *F (df = 5,14)* | | *213.18(P<0.001)* | *6.21(P<0.001)* | *6.68(P<0.001)* |
| | 2 | 15 | IMM | 100 ± 0.00[a] | 62.26 ± 8.88[a] | 0.23 ± 0.01[b] |
| | | | WXM | 100 ± 0.00[a] | 60.39 ± 4.37[a] | 0.26 ± 0.02[a] |
| | | 30 | IMM | 96.49 ±3.51[a] | 43.83 ± 2.06[ab] | 0.23 ± 0.03[b] |
| | | | WXM | 95.65 ± 4.35[ab] | 53.94 ± 5.26[a] | 0.24 ± 0.01[ab] |
| | | 45 | IMM | 99.33± 0.67[a] | 45.74 ± 4.13[ab] | 0.21± 0.02[c] |
| | | | WXM | 78.57 ± 6.00[b] | 46.67 ± 3.3[ab] | 0.20 ± 0.02[c] |
| | | 60 | IMM | 33.33 ±5.21[c] | 15.65 ± 2.49[c] | 0.20 ± 0.03[c] |
| | | | WXM | 45.00 ± 2.89[c] | 24.48 ± 1.46[bc] | 0.20 ± 0.02[c] |
| | | *F (df = 7,16)* | | *56.45(P<0.001)* | *13.05(P<0.001)* | *11.71(P<0.001)* |

Within a column for each temperature and age, means followed by the same letter do not differ significantly; HSD test at 0.05.

*mellonella* (F = 531.93; df = 3,8; P<0.001) (Table 3). In this series of tests, we saw that developmental periods of the parasitoid were insignificantly influenced by species (F = 1.79; df = 1,16; P = 0.20) and interaction species*temperature (F = 1.21; df = 3,16; P = 0.33).Temperature had a significant effect in *T. evanescens* adult body length (F = 327.86; df = 7,41; P = 0.001) for both

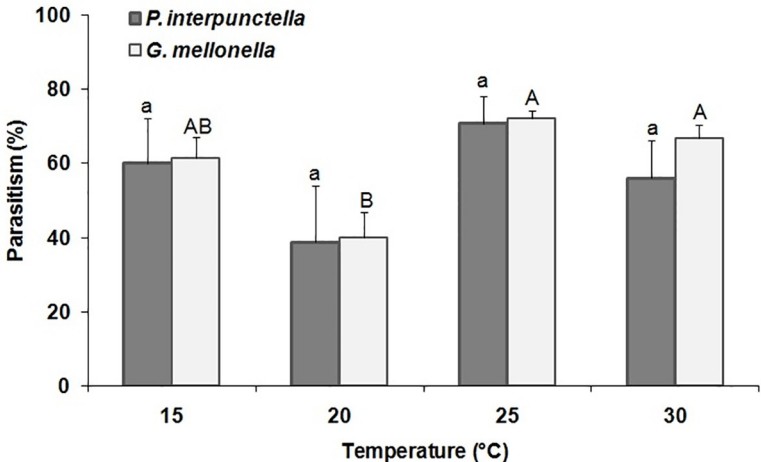

**Fig 3. Mean percentage (% ±SE) of parasitism by *T. evanescens* in *P. interpunctella* and *G. mellonella* eggs on different temperatures (within each species, bars followed by the same letters are not significantly different; HSD test at 0.05; lowercase letters for *P. interpunctella*, uppercase letters for *G. mellonella*).**

species (Table 3). In most combinations, adult body length was higher in the individuals that were developed from *G. mellonella*, as compared with the respective figures of *P. interpunctella*. The body length varied significantly among temperatures (F = 171.68; df = 3,41; P = 0.001) and between the species (F = 1752.03; df = 1,41; P = 0.001). There were also significant variations (F = 9.32; df = 3,41; P = 0.001)in the interaction of species*temperature for the adult body length. Finally, the highest mortality levels were recorded in *T. evanescens* reared at 20°C for either host species, as indicated by the higher *k*-values (Table 3).

## Discussion

Egg parasitoids have to be efficient in locating hosts as the life span of the egg stage is generally short under normal conditions. Moreover, previous studies indicate that old eggs are less suitable as hosts as they adversely affect several parasitoid biological parameters, such as percentage of parasitism, developmental time [37], adult emergence [38], body size and sex ratio [39].

**Table 3. Effect of temperature (mean ±SE) on the biological traits of *T. evanescens* reared on eggs of *P. interpunctella* (IMM) and *G. mellonella* (WXM).**

| Species | Temperature (°C) | Adult emergence (%) | Female progeny (%) | Developmental periods (d) | Adult body length (mm) | *k*- values |
|---|---|---|---|---|---|---|
| IMM | 15 | 86.92 ± 4.95[a] | 67.14 ± 10.81[a] | 23.67 ± 0.88[a] | 0.24 ± 0.01[a] | 0.70 ± 0.29 |
| | 20 | 80.22 ± 9.92[a] | 65.56 ± 8.69[a] | 19.33 ± 0.33[b] | 0.25 ± 0.01[a] | 1.49 ± 0.52 |
| | 25 | 94.91 ± 2.89[a] | 33.60 ± 11.52[a] | 9.37 ± 0.19[c] | 0.23 ± 0.0[a] | 0.41 ± 0.08 |
| | 30 | 92.59 ± 7.42[a] | 42.47 ± 2.77[a] | 7.33 ± 0.21[c] | 0.23 ± 0.01[a] | 0.70 ± 0.29 |
| *F (df = 3,8)* | | *1.31(P = 0.34)* | *3.38 (P = 0.07)* | *251.41(P<0.001)* | *1.81(P = 0.16)* | |
| WXM | 15 | 89.29 ± 1.64[ab] | 20.51 ± 12.86[b] | 24.67 ± 0.67[a] | 0.27 ± 0.01[a] | 0.61 ± 0.09 |
| | 20 | 88.21 ± 6.05[b] | 37.72 ± 13.95[ab] | 20.33 ± 0.51[b] | 0.25 ± 0.01[a] | 1.08 ± 0.13 |
| | 25 | 100 ± 0.00[a] | 54.84 ± 11.10[ab] | 9.00 ± 0.08[c] | 0.26 ± 0.01[a] | 0.33 ± 0.03 |
| | 30 | 98.04 ± 1.96[a] | 66.76 ± 5.56[a] | 7.00 ± 0.08[d] | 0.25 ± 0.01[a] | 0.43 ± 0.05 |
| *F (df = 3,8)* | | *7.90 (P<0.008)* | *4.14 (P<0.04)* | *531.93(P<0.001)* | *2.22 (P = 0.09)* | |

Within a column for each species, means with the same letter do not differ significantly; where no letters exist, no. significant differences were noted; HSD test at 0.05.

These effects may be due to changes in the chemical composition as nutrients that are gradually consumed by the host embryo and change, or there are changes in the physical characteristics of the chorion, which becomes more rigid as the egg ages [40,41]. In this context, it is well established that preference for hosts towards a particular host age group may enhance the fitness of egg parasitoids [42,43]. Thus, parasitoids might make use of the physical characteristics and/or chemicals in and on the surface of eggs, which drastically change with age, as critical cues indicating the suitability of hosts. Nevertheless, our results demonstrate that egg age may not be of critical importance, at least in the case of the species range tested here, and can be moderated by other factors, such as the host species and previous exposure to low temperatures. In this context, the differences between the age groups utilized here were not very diverse, in terms of rates of parasitism.

We have found an interesting interaction between egg age and storage period. As a general principle, we saw that the percentage of parasitism was gradually decreased with the increase of the cold storage period. However, this decrease exhibited similar trends for both parasitoids only in the case of 1d-old eggs. In contrast, for the 2 d-old eggs *G. mellonella* performed better as a host at 4°C, providing a considerable percentage of parasitized eggs, even at the longer storage period (60 d). It is generally considered that *G. mellonella* eggs take longer to hatch, as compared with those of *P. interpunctella* [44,45], although comparative studies are not available towards this direction. Hence, the difference between the two egg age groups for *G. mellonella* can be narrower as compared with the respective figures of *P. interpunctella*, which indicates that *G. mellonella* can be more suitable as a host for *T. evanescens*, in terms of more gradual egg "maturation". In an earlier comparison of these two species as hosts of the larval ectoparasitoid *Harbobracon hebetor* (Say) (Hymenoptera: Braconidae), Hasan et al. [29] considered *G. mellonella* as a superior host, due to certain larval characteristics, such as size and longevity. Paradoxically, our data illustrate that 2 d-old eggs of this host species maintain a good level of suitability for parasitism by *T. evanescens*, for a long period of time, which means that 2 d-old eggs are less affected by storage periods as compared with those of *P. interpunctella*. Nevertheless, morphological variations may drastically contribute to egg suitability for parasitism. For instance, the egg parasitoid *Telenomus remus* Nixon (Hymenoptera: Scelionidae) is able to parasitize the eggs in the basal layers of *Spodoptera* spp. (Lepidoptera: Noctuidae) egg masses, even when they are covered by moth scales, much more effectively than *Trichogramma* spp. [46], suggesting that the latter is more "selective" regarding egg preference.

In contrast with storage at 4°C, storage at 9°C differentiated the parasitism rates in a more uniform way for both host species. However, even in this temperature, parasitism rates were higher at *G. mellonella* eggs, as compared with *P. interpunctella* eggs. Storage of *G. mellonella* at this temperature provided very high percentages of parasitism up to 30 d; in fact, storage for 15 and 30 d provided even higher percentages than untreated eggs (0 d). For the larval endoparasitoid *Venturia canescens* (Gravenhorst) (Hymenoptera: Ichneumonidae), Eliopoulos et al. [47] noted that there was a critical egg maturation stage, which was directly related with viability and progeny production capacity. For the same species, Andreadis et al. [48] found that cold tolerance was negatively correlated with parasitoid age. We are unaware if egg exposure to low temperatures increases their suitability for parasitism, in the same way that larval irradiation positively affects parasitism in moth larvae by *H. hebetor* [49], but this hypothesis needs additional investigation. Considering the overall data for this series of bioassays, we assume that older eggs were more susceptible to low temperatures as compared to young eggs.

The combined use of both host species generally provided higher parasitism rates, which, paradoxically, did not always yield in higher parasitoid emergence rates. Adult parasitoid emergence in the combined use of both host species, was reduced in eggs that were 2 and 3 d-old, as compared with the use of one single host species. Spatial segregation of hosts may result

in different parasitism rates, resulting in host preference patterns and parasitoid aggregation in specific host groups [50]. Adaptation and counter-adaptation of a parasitoid in different hosts that can coexist locally is directly related with host rate advantage [51]. Practically, the simultaneous utilization of more than one host species may offer specific benefits to the overall parasitoid rearing technique, as parasitism may be shifted to an alternative host, if there is a breakdown in the case of the superior host species, e.g. change of egg suitability.

In contrast with egg age and previous exposure to cold, parasitism rates were similar on both host species, regardless of the temperature level for the rearing of the parasitoids, at least at the range of temperatures examined here. Moreover, despite the fact that we observed a reduction in parasitism at 20˚C, the increase of temperature beyond that level further increased the parasitism. In a similar series of tests with *Trichogramma turkestanica* Meyer, reared of eggs of the Mediterranean flour moth, *Ephestia kuehniella* Zeller (Lepidoptera: Pyralidae), Hansen and Jensen [52] found similar temperature-dependent parasitism trends. Still, different species of *Trichogramma* exhibit dissimilar developmental and parasitism rates in relation with temperature, but there are species of this genus for which population growth is benefited by the increase of temperature [53]. Consequently, our data demonstrate that maximum parasitism can be achieved at temperatures that are higher than 20˚C, which should be taken into account in *T. evanescens* rearing protocols, or enhancement of the parasitoid progeny production when using eggs that have been previously exposed to low temperatures.

Paradoxically, exposure of eggs to low temperatures changed the female ratio, in a different way for each of the host species. In some of the combinations tested, the female ratio was higher in untreated (not exposed to cold) eggs, which partially explains the higher parasitism rates that were recorded in these combinations, despite the fact that the highest female ratio was not always correlated with higher rates. Moreover, for 1 d-eggs that had been exposed to either 4 or 9˚C, we observed a gradual increase of the female ratio, while the reverse was noted for 2 d-old eggs. It has been well established that the female ratio of some species of *Trichogramma* can be significantly altered at different temperatures, resulting in different parasitism and longevity rates [53]. In contrast with the female ratio, adult body length was decreased with the increase of the exposure interval in low temperatures. "Better eggs", in terms of size and nutrients, are known to provide larger parasitoids, but the size of parasitoids may not necessarily result in a better parasitism performance [54].

## Conclusions

Our work demonstrated that storage of eggs for a certain internal can be used with success for rearing protocols of *T. evanescens*. Between the two hosts used here, *G. melonella* was more suitable for parasitoid rearing, mainly due to the fact that this species provides eggs that are more suitable for parasitism when exposed at low temperatures. Maintaining moth eggs at low temperatures has been proposed as a means to control parasitoid rearing, and to provide large numbers of parasitoids whenever it is needed, utilizing both moth species. Having alternative host species in egg parasitoid rearings can prevent unexpected turnovers when there are problems with the rearing of the basic host. In this effort, 2 d-old eggs can be more suitable than 1 d-old eggs, and this age could be proposed for selection to increase parasitoid performance, especially when maintained at 9˚C.

## Acknowledgments

We thank to the Bangladesh Agriculture Research Institute (BARI), Gazipur, Bangladesh for providing the parasitoids and hosts. The authors are grateful to the Department of Zoology,

Rajshahi University for extending the laboratory facilities. We also thank the two anonymous reviewers that provided useful comments on the original manuscript.

## Author Contributions

**Conceptualization:** Saiful Islam, Christos G. Athanassiou, Mahbub Hasan.

**Data curation:** Aslam Haque, Abdul Bari.

**Formal analysis:** Aslam Haque, Mahbub Hasan.

**Investigation:** Aslam Haque, Abdul Bari, Akhtar Hossain.

**Writing – original draft:** Mahbub Hasan.

**Writing – review & editing:** Christos G. Athanassiou.

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
