## [Decision Letter · Decision Letter 0]

10 Mar 2021

PONE-D-21-01652

Enhancing the quality control for mass rearing of Trichogramma evanescens Westwood reared on the Indian meal moth Plodia interpunctella (Hübner) and the wax mothGalleria mellonella L.

PLOS ONE

Dear Dr. Hasan,

Thank you for submitting your manuscript to PLOS ONE. After careful consideration, we feel that it has merit but does not fully meet PLOS ONE’s publication criteria as it currently stands. Therefore, we invite you to submit a revised version of the manuscript that addresses the points raised during the review process by the two reviewers.

We look forward to receiving your revised manuscript.

Kind regards,

Nicolas Desneux

Academic Editor

PLOS ONE

Journal Requirements:

Reviewers' comments:

Reviewer's Responses to Questions

**Comments to the Author**

1. Is the manuscript technically sound, and do the data support the conclusions?

Reviewer #1: Yes

Reviewer #2: Partly

2. Has the statistical analysis been performed appropriately and rigorously? 

Reviewer #1: No

Reviewer #2: No

3. Have the authors made all data underlying the findings in their manuscript fully available?

Reviewer #1: Yes

Reviewer #2: Yes

4. Is the manuscript presented in an intelligible fashion and written in standard English?

Reviewer #1: Yes

Reviewer #2: Yes

5. Review Comments to the Author

Reviewer #1: The egg parasitoid Trichogramma evanescens is widely distributed throughout the world, and considered as an efficient biological control agent against many lepidopteran pests. In this study, the authors investigated effects of host species, egg age, cold storage and temperature on the quality control in mass rearing of the parasitoid. The results provide some valuable information on instructing mass production of T. evanescens with commonly intermediate hosts, the Indian meal moth, Plodia interpunctella and the wax moth, Galleria mellonella. Through checking this manuscript, the experiment design and writing is ok, the results also can support the conclusion. However, there are serious defects in data analyses. In addition, too many old references were cited in this paper. Some specific comments see the followings.

Lines 57-59: For the references of 1-7, some are too old, the authors may replace them with some new publications. Some references are recommended as followed.

Wang Y, et al., 2020. Manually-extracted unfertilized eggs of Chinese oak silkworm, Antheraea pernyi, enhance mass production of Trichogramma parasitoids. Entomologia Generalis, 40:397-406.

Zhang JJ, et al., 2018. Advantages of diapause in Trichogramma dendrolimi mass production via eggs of the Chinese silkworm, Antheraea pernyi. Pest Management Science, 74: 959-965.

Lines 60-62: The same comments to lines 57-59. Some references are recommended as followed.

Hou YY, et al., 2018. Effect of oriental armyworm Mythimna separata egg age on the parasitism and host suitability for five Trichogramma species. Journal of Pest Science, 91: 1181–1189.

Zhang JJ, et al., 2014. Effects of host-egg ages on host selection and suitability of four Chinese Trichogramma species, egg parasitoids of the rice striped stem borer, Chilo suppressalis. BioControl. 59: 159-166.

Lines 120-122: Change T.evanescens and P.interpunctella to T. evanescens and P. interpunctella.

Lines 130- 135: How long the host eggs were expose to parasitoids?

Lines 178-179: these descriptions are not matched with the experimental contents.

Lines 181-187: The data analyses are described too simple. In this paper, the authors conducted three experiments, in which different factors have been considered. The authors should describe ANOVA with multi-factors in detail.

Lines 191-192: I’m confused on df=5, 30, why the treatment freedom is 5?

Lines 194: why the treatment freedom is 3?

Lines 195-196: Total treatments are 12, why the treatment freedom is 13? All the analysis data given in the text should be checked, particular for the freedom data!

Line 207: change dold to d old.

Line 212: change 4 d to 14 d.

Lines 225-228: I can’t find the parasitism data in Table 2.

Lines 277-278: only two species, why the treatment freedom is 7? Carefully check all the freedom data in the results.

Lines 634-637: For Table 1, I’m confused on all difference letters given. It is too chaos about the sequence of letters listed. Seemingly the a, b, c..... were given randomly! In addition, in the method, 25 eggs per card, replicate 5 times, why 75 eggs were given in the table? In the table, the host eggs of combined species were given, but I can’t find the method description on it!

I also don’t agree to use Female proportion, it is better to change it to percentage of female progeny (%)

Lines 640-644: the same comments to Table 1.

Lines 646-649: the same comments to Table 1.

For Figure 2, change Cold storage period (%) to Cold storage period (d)

Reviewer #2: Dear Authors

Despite a big amount of data was collect and the topic is relavant some weakness was found in your MS.

1) The hypothesis is not very clear.2

2) There isn’t correlation between title, hypothesis and conclusion.

3) Data transformation is not recommended. see the article: “The arcsine is asinine: the analysis of proportions in ecology. Warton and Hui. Ecology. V. 92. 2011.”

4) When you are expressing mass rearing are you are talking about which amount of eggs? 1 Kg per day? This should be considered in order to address a more realistic scale and fit better your finds with the biofactories reality, and also to give for you reader the real idea of what is a mass rearing.

5) In my opinion your data is very useful and interesting, but no great innovation is presented.

6. PLOS authors have the option to publish the peer review history of their article (what does this mean?). If published, this will include your full peer review and any attached files.

Reviewer #1: No

Reviewer #2: No

---

## [Author Response · Author response to Decision Letter 0]

21 Mar 2021

Response to Reviewers Comment

Reviewer #1: The egg parasitoid Trichogramma evanescens is widely distributed throughout the world, and considered as an efficient biological control agent against many lepidopteran pests. In this study, the authors investigated effects of host species, egg age, cold storage and temperature on the quality control in mass rearing of the parasitoid. The results provide some valuable information on instructing mass production of T. evanescens with commonly intermediate hosts, the Indian meal moth, Plodia interpunctella and the wax moth, Galleria mellonella. Through checking this manuscript, the experiment design and writing is ok, the results also can support the conclusion. However, there are serious defects in data analyses. In addition, too many old references were cited in this paper. Some specific comments see the followings.

Lines 57-59: For the references of 1-7, some are too old, the authors may replace them with some new publications. Some references are recommended as followed.

Response: References have been included as well as revised as suggested, see line 443, 446

Wang Y, et al., 2020. Manually-extracted unfertilized eggs of Chinese oak silkworm, Antheraeapernyi, enhance mass production of Trichogramma parasitoids. EntomologiaGeneralis, 40:397-406.

Zhang JJ, Zhang X, Zang LS, , Du WM, Hou YY, Ruan CC, et al.2018. Advantages of diapause in Trichogrammadendrolimi mass production via eggs of the Chinese silkworm, Antheraeapernyi. Pest Manag. Sci., 2018; 74: 959-965.

Lines 60-62: The same comments to lines 57-59. Some references are recommended as followed.

Response: References have been included as well as revised as suggested, see line 453, 463

Hou YY, Yang X, Zang LS, Zhang C, Monticelli LS, Desneux N. Effect of oriental armyworm Mythimnaseparata egg age on the parasitism and host suitability for five Trichogramma species. J Pest Sci.2018; 91: 1181–1189.

Zhang JJ, Ren BZ, Yuan XH, Zang LS, Ruan CC, Sun GZ, Shao XW, Effects of host-egg ages on host selection and suitability of four Chinese Trichogramma species, egg parasitoids of the rice striped stem borer. Chilosuppressalis.BioControl, 2014; 59:159–166

Lines 120-122: Change T.evanescens and P.interpunctella to T. evanescens and P. interpunctella.

Response: Has been corrected, see line 122

Lines 130- 135: How long the host eggs were expose to parasitoids?

Response: Has been corrected, see line 135

Lines 178-179: these descriptions are not matched with the experimental contents.

Response: Has been corrected, see line 181

Lines 181-187: The data analyses are described too simple. In this paper, the authors conducted three experiments, in which different factors have been considered. The authors should describe ANOVA with multi-factors in detail.

Response: The ANOVA with multi-factors have been done in details, see line 188

Lines 191-192: I’m confused on df=5, 30, why the treatment freedom is 5?

Response: Have rechecked and corrected, see line 193-194

Lines 194: why the treatment freedom is 3?

Response: Have been rechecked and corrected, see line 195-196. We have actually two species including two combinations.

Lines 195-196: Total treatments are 12, why the treatment freedom is 13? All the analysis data given in the text should be checked, particular for the freedom data!

Response: Have been rechecked and corrected, see line 202

Line 207: change dold to d old.

Response: Has been corrected, see line 212

Line 212: change 4 d to 14 d.

Response: Has been corrected, see line 219

Lines 225-228: I can’t find the parasitism data in Table 2.

Response: Has been corrected, see line 237

Lines 277-278: only two species, why the treatment freedom is 7? Carefully check all the freedom data in the results.

Response: Has been corrected, see line 294

Lines 634-637: For Table 1, I’m confused on all difference letters given. It is too chaos about the sequence of letters listed. Seemingly the a, b, c..... were given randomly! In addition, in the method, 25 eggs per card, replicate 5 times, why 75 eggs were given in the table? In the table, the host eggs of combined species were given, but I can’t find the method description on it!

Response: Has been corrected in the method as three replications, see line 140.

I also don’t agree to use Female proportion, it is better to change it to percentage of female progeny (%)

Response: Has been corrected as well as the Female proportion, which has been changed into percentage of female progeny (%). see table 1 and 2.

Lines 640-644: the same comments to Table 1.

Response: Has been corrected, see Table 1

Lines 646-649: the same comments to Table 1.

Response: Has been corrected, see table 1

For Figure 2, change Cold storage period (%) to Cold storage period (d)

Response: Caption unit has been changed, see caption of x axis in Fig 2

Reviewer #2:

Dear Authors

Despite a big amount of data was collect and the topic is relevant some weakness was found in your MS.

1) The hypothesis is not very clear.

Response: Thank you for this comment. We agree that the theory behind the aim of the current work needs additional clarification. For this purpose, we have included an additional text at the “aims” part of the Introduction, see lines 97-99

2) There isn’t correlation between title, hypothesis and conclusion.

Response: Thanks for this comment. Indeed, the title was too generic; hence we have revised the title as: “Cold storage-mediated rearing of Trichogramma evanescens Westwood on eggs of Plodia interpunctella (Hübner) and Galleria mellonella L.”, which represents more accurately the core idea of this study (see title in the revised ms).

3) Data transformation is not recommended. see the article: “The arcsine is asinine: the analysis of proportions in ecology. Warton and Hui. Ecology. V. 92. 2011.”

Response: We have followed a standardized procedure. At first the assumptions of homogeneity and homoscedasticity are addressed, through different initial screening tests, such as Levene, O’Brien, Bartlett etc. When these assumptions are not met, then data transformation is performed, and if this is successful, then parametric tests are followed. This standard procedure is followed in numerous papers, on an extremely large variety of topics. We are aware of the Warton and Hui paper, that illustrates that arcsine transformation is non-desirable in either on binomial or non-binomial data sets. Nevertheless, there are several studies that followed Warton and Hui work, that have shown merit in this type of transformation, indicatively, Martin et al. (2013), PNAS, and especially Lin and Hu (2020), Health Science Reports, that show some merit in using arcsine. In this context, we would rather to maintain the transformation approach used here.

4) When you are expressing mass rearing are you are talking about which amount of eggs? 1 Kg per day? This should be considered in order to address a more realistic scale and fit better your finds with the biofactories reality, and also to give for you reader the real idea of what is a mass rearing.

Response: We agree with this comment. The long-term aim of the current series of tests is to provide some additional data that can be used in mass rearing protocols. Apparently, the idea here is to test these parameters at the lab scale. Therefore, as suggested, we have removed the “mass rearing” notifications from most parts of the text, and we have kept this expression in few occasions at the introduction section, to illustrate the long-term aim of our work (see lines 96). 

5) In my opinion your data is very useful and interesting, but no great innovation is presented.

Response: Based on what is mentioned at the introduction section, and also at the conclusions, there is still inadequate data on the effect of long-term storage of moth eggs towards rearing of Trichogramma, especially in the case of the host species tested here. We consider that this can be a noticeable innovation, and can be further used in rearing protocols of moth egg parasitoids, as an “on-off” procedure. Also, having data for more than one host species may be useful in the case of turnovers, when the rearing of the basic host species is problematic. For this purpose, we have added a separate notification at the Conclusions, see lines 405.

---

## [Decision Letter · Decision Letter 1]

5 May 2021

PONE-D-21-01652R1

Cold storage-mediated rearing of Trichogramma evanescens Westwood on eggs of Plodia interpunctella (Hübner) and Galleria mellonella L.

PLOS ONE

Dear Dr. Hasan,

Thank you for submitting your manuscript to PLOS ONE. After careful consideration, we feel that it has merit but does not fully meet PLOS ONE’s publication criteria as it currently stands. Therefore, we invite you to submit a revised version of the manuscript that addresses the points raised during the review process.

We look forward to receiving your revised manuscript.

Kind regards,

Nicolas Desneux

Academic Editor

PLOS ONE

Additional Editor Comments (if provided):

The statistical analyses are not properly presented and therefore their validity is questionable. Because of this the paper can not be accepted for publication in Plos ONE yet. The authors should likely use a GLM analysis instead of doing transformation on datasets. At minimum, authors should present the statistical analyses in a better way (e.g see the comments from reviewer 1).

Reviewers' comments:

Reviewer's Responses to Questions

**Comments to the Author**

1. If the authors have adequately addressed your comments raised in a previous round of review and you feel that this manuscript is now acceptable for publication, you may indicate that here to bypass the “Comments to the Author” section, enter your conflict of interest statement in the “Confidential to Editor” section, and submit your "Accept" recommendation.

Reviewer #1: (No Response)

Reviewer #2: All comments have been addressed

2. Is the manuscript technically sound, and do the data support the conclusions?

Reviewer #1: Yes

Reviewer #2: Yes

3. Has the statistical analysis been performed appropriately and rigorously? 

Reviewer #1: No

Reviewer #2: No

4. Have the authors made all data underlying the findings in their manuscript fully available?

Reviewer #1: Yes

Reviewer #2: Yes

5. Is the manuscript presented in an intelligible fashion and written in standard English?

Reviewer #1: Yes

Reviewer #2: Yes

6. Review Comments to the Author

Reviewer #1: The qualtiy of updated manuscript has been greatly improved based on the revision suggestions, but there are still serious problems on statistical analyses.

1. lines 184-186: The authors did not give the description in detail. For different trials, seperately describe the statistical methods using ANOVA with multi-factors, what factors have been given for each trial.

2. Table 1: I'm still confused on some difference letters given in table, for example, for the results of adult emergence on 1 day age egg, the data of 87.50 and 91.67 with 'a' and 'ab', respectively. It should give the data: 91.67a, 87.5 ab. Carefully check all the statistical analysis results for Table 1, 2, and 3, then give the right letters for various treatments.

3. Fig. 1: I'm confused on all difference letters given on the bars. It is not clear to note "within eahc egg age, bars followed by the same letters are not significantly different; HSD test at 0.05". Checking Fig. 1, I'm confused on the indication. How you do the comparisions for two species at the same egg age, both togher or seperately? Similarly, the letters on the bars are also given in disorder, check all the statistical results carefully, then give the correct letters for various treatment.

4. Fig. 2: same to Fig. 1. particularly note: for 1 day old egg, at 15 d, why give the letter 'ad' for P. interpunctella?

Reviewer #2: Dear author

Thank you for reply all comments. You presented a very good argumentation, it was good to read it.

7. PLOS authors have the option to publish the peer review history of their article (what does this mean?). If published, this will include your full peer review and any attached files.

Reviewer #1: No

Reviewer #2: No

---

## [Author Response · Author response to Decision Letter 1]

13 May 2021

Response to Reviewers

Reviewer #1: The quality of updated manuscript has been greatly improved based on the revision suggestions, but there are still serious problems on statistical analyses.

Response: Thanks for the appreciation

1. lines 184-186: The authors did not give the description in detail. For different trials, separately describe the statistical methods using ANOVA with multi-factors, what factors have been given for each trial.

Response: Revised the statistical methods as suggested. Pls see lines 182-190

2. Table 1: I'm still confused on some difference letters given in table, for example, for the results of adult emergence on 1 day age egg, the data of 87.50 and 91.67 with 'a' and 'ab', respectively. It should give the data: 91.67a, 87.5 ab. Carefully check all the statistical analysis results for Table 1, 2, and 3, then give the right letters for various treatments.

Response: The results for comparison letters presented in Tables 1, 2, 3 are checked and corrected accordingly

3. Fig. 1: I'm confused on all difference letters given on the bars. It is not clear to note "within each egg age, bars followed by the same letters are not significantly different; HSD test at 0.05". Checking Fig. 1, I'm confused on the indication. How you do the comparisions for two species at the same egg age, both togher or seperately? Similarly, the letters on the bars are also given in disorder, check all the statistical results carefully, then give the correct letters for various treatment.

Response: The results particularly for the comparison letters presented in Fig 1 are checked and corrected accordingly

4. Fig. 2: same to Fig. 1. particularly note: for 1 day old egg, at 15 d, why give the letter 'ad' for P. interpunctella?

Response: The results particularly comparison letters presented in Fig 2 are checked and corrected accordingly

Reviewer #2: Dear author

Thank you for reply all comments. You presented a very good argumentation; it was good to read it.

Response: Thanks for the appreciation

---

## [Editor Report · Decision Letter 2]

2 Jun 2021

Cold storage-mediated rearing of Trichogramma evanescens Westwood on eggs of Plodia interpunctella (Hübner) and Galleria mellonella L.

PONE-D-21-01652R2

Dear Dr. Hasan,

We’re pleased to inform you that your manuscript has been judged scientifically suitable for publication and will be formally accepted for publication once it meets all outstanding technical requirements.

Kind regards,

Nicolas Desneux

Academic Editor

PLOS ONE

---

## [Editor Report · Acceptance letter]

4 Jun 2021

PONE-D-21-01652R2 

Cold storage-mediated rearing of *Trichogramma evanescens* Westwood on eggs of *Plodia interpunctella* (Hübner) and *Galleria mellonella* L. 

Dear Dr. Hasan:

I'm pleased to inform you that your manuscript has been deemed suitable for publication in PLOS ONE. Congratulations! Your manuscript is now with our production department. 

Kind regards, 

on behalf of

Dr. Nicolas Desneux 

Academic Editor

PLOS ONE